# A Diet Enriched with *Lacticaseibacillus rhamnosus* HN001 and Milk Fat Globule Membrane Alters the Gut Microbiota and Decreases Amygdala GABA a Receptor Expression in Stress-Sensitive Rats

**DOI:** 10.3390/ijms241310433

**Published:** 2023-06-21

**Authors:** Julie E. Dalziel, Gosia Zobel, Hilary Dewhurst, Charlotte Hurst, Trent Olson, Raquel Rodriguez-Sanchez, Louise Mace, Nabil Parkar, Caroline Thum, Rina Hannaford, Karl Fraser, Alastair MacGibbon, Shalome A. Bassett, James Dekker, Rachel C. Anderson, Wayne Young

**Affiliations:** 1Smart Foods & Bioproducts, AgResearch, Palmerston North 4442, New Zealand; hilary.dewhurst@agresearch.co.nz (H.D.);; 2Ethical Agriculture, AgResearch, Hamilton 3240, New Zealand; gosia.zobel@agresearch.co.nz (G.Z.); raquel.rodriguez@agresearch.co.nz (R.R.-S.); 3Digital Agriculture, AgResearch, Palmerston North 4442, New Zealand; 4Fonterra Research and Development Centre Co., Ltd., Palmerston North 4442, New Zealand

**Keywords:** gut–brain axis, neurotransmitter, behaviour, lipid, stress, anxiety, depression

## Abstract

Brain signalling pathways involved in subclinical anxiety and depressed mood can be modulated via the gut brain axis (GBA), providing the potential for diet and dietary components to affect mood. We investigated behavioural, physiological and gut microbiome responses to the *Lacticaseibacillus rhamnosus* strain HN001 (LactoB HN001™), which has been shown to reduce postpartum anxiety and depression, and a milk fat globule membrane-enriched product, Lipid 70 (Surestart^TM^ MFGM Lipid 70), which has been implicated in memory in stress-susceptible Wistar Kyoto rats. We examined behaviour in the open field, elevated plus maze and novel object recognition tests in conjunction with the expression of host genes in neuro-signalling pathways, and we also assessed brain lipidomics. Treatment-induced alterations in the caecal microbiome and short-chain fatty acid (SCFA) profiles were also assessed. Neither ingredient induced behavioural changes or altered the brain lipidome (separately or when combined). However, with regard to brain gene expression, the *L. rhamnosus* HN001 + Lipid 70 combination produced a synergistic effect, reducing GABAA subunit expression in the amygdala (*Gabre*, *Gat3*, *Gabrg1*) and hippocampus (*Gabrd*). Treatment with *L. rhamnosus* HN001 alone altered expression of the metabotropic glutamate receptor (*Grm4*) in the amygdala but produced only minor changes in gut microbiota composition. In contrast, Lipid 70 alone did not alter brain gene expression but produced a significant shift in the gut microbiota profile. Under the conditions used, there was no observed effect on rat behaviour for the ingredient combination. However, the enhancement of brain gene expression by *L. rhamnosus* HN001 + Lipid 70 implicates synergistic actions on region-specific neural pathways associated with fear, anxiety, depression and memory. A significant shift in the gut microbiota profile also occurred that was mainly attributable to Lipid 70.

## 1. Introduction

Stress can be considered a reaction to pressurising and challenging circumstances, both physiologically and mentally, with multiple underlying factors. Stress and resilience to stress are now understood to involve peripheral and central bi-directional systems between the gut and the brain, termed the gut–brain axis (GBA) [1]. Subclinical anxiety and depressed mood in mentally healthy individuals have the potential to be modulated by food ingredients that influence GBA signalling and help improve brain health [2,3,4]. An important modulator of the ultimate impact of foods on health and wellbeing is the gut microbiome. The impact of the gut microbiome on brain function has been well documented, even if the precise mechanisms themselves are not entirely understood. The complex communication system between the microbiome and the peripheral and central nervous systems is often altered in psychiatric disorders and upon exposure to stress and anxiety. In rodent studies, transfer of the microbiome from human donors suffering from autism spectrum disorders can lead to behavioural abnormalities and altered neuronal activity [5], illustrating the potentially causal relationship of the gut microbiome with brain function. While the processes driving the GBA are undoubtably complex, there is evidence that they can be modified by diet [6]. One example of a food ingredient that might improve brain health is probiotic bacteria. The idea that probiotics may offer an effective adjunct therapy for people with depression was first hypothesised in 2005 [7]. Since then, the use of probiotics to help reduce depression symptoms has been rapidly growing, but with mixed success. Some studies report reductions in anxiety and depression symptoms in participants receiving probiotics [8,9,10,11,12,13], whereas other studies report no beneficial effects [14,15]. This indicates that strain specificity in the development of probiotic treatments may be key to their effectiveness. The *Lacticaseibacillus rhamnosus* strain HN001 (LactoB HN001™) has been shown to reduce symptoms of postpartum anxiety and depression when taken during and after pregnancy [16]. Its ability to improve gut-barrier integrity and immune function in mice [17,18] suggests that it may potentially also affect GBA pathways.

In this study, we wanted to investigate the actions of combined ingredients on brain function; in particular, a probiotic with milk fat globule membrane (MFGM). MFGM contains complex lipids, including gangliosides and sphingolipids, that are important in neuronal cell membrane structure, metabolism and neurotransmission. Lipid-enriched milk fractions improve memory in young rats [19] and, in particular, spatial memory rather than recognition memory in a rat model of early life stress [20]. In human studies, milk lipids improve working memory during acute stress [21] and reaction time under psychosocial stress [22]. Although MFGM alone does not affect recognition memory or brain structural lipids in piglets [23], when provided in combination with prebiotics and lactoferrin, it improves spatial memory and structural neurodevelopment [24]. This suggests that providing MFGM in combination with other brain-modulating ingredients such as probiotics might improve performance in brain function tests.

The aim of this research was to investigate whether a combination of *L. rhamnosus* HN001 and Lipid 70 (Surestart^TM^ MFGM Lipid 70) would be more effective than either ingredient separately at altering behaviour and brain markers associated with anxiety, depression and memory. We used the Wistar Kyoto (WKY) rat strain because it is genetically predisposed to anxiety and depression [25] and has heightened reactivity to acute and chronic stress compared with normo-sensitive rat strains [26,27]. In terms of brain neurochemistry, alterations in monoamines, GABA and glutamate pathways and a serotonin deficit are hallmarks of the WKY phenotype [27,28]. WKY gastrointestinal (GI) tract function is characterised by gastroparesis, bile-acid lipid-metabolism-associated dysbiosis [29], visceral hypersensitivity [30,31], altered colonic pathology [32] and corticotropin-releasing factor (CRF) receptor expression [33]. Furthermore, the microbiota profile of WKY rats is affected by acute stress [26,30,31] in a manner that correlates with their brain lipid profile, suggesting that GBA pathways are disrupted and that the activity of microbiota influences these pathways [26].

We assessed the effect of *L. rhamnosus* HN001- and Lipid 70-supplemented diets, separately and together, on rat behaviour in the open field, elevated plus maze and novel object recognition tests in conjunction with measuring brain region-specific differences in gene expression in neuro-signalling pathways and associated lipidomics. Treatment-induced alterations in the caecal microbiome and corresponding changes in short-chain fatty acid profiles (SCFA) were also determined.

## 2. Results

### 2.1. Animal Metrics

Animal weight increased from a pre-treatment mean of 388 ± 5 g (n = 63) (*p* < 2 × 10^−16^) by 8.5% per week over the course of the study. However, this rate of increase was not significantly different among treatment groups. Because food intake was measured every 2–4 days, it is shown as cumulative intake (Appendix A). Food intake did not differ among treatment groups. The *L. rhamnosus* HN001 intake in drinking water was ~30 mL/rat/day, which equated to a dose of 1.5 × 10^9^ CFU/rat/day.

### 2.2. Behaviour Tests

No behavioural differences were detected due to treatment in any of the assessment measures across the open field test (OFT), indicative of anxiety and exploration; the elevated plus maze (EPM) test, indicative of anxiety; and the novel object recognition (NOR) test for recognition memory (Appendix A). Both the treatment and control groups were timid in their exploration of the open areas and objects.

### 2.3. Brain Gene Expression

The results showed differential gene expression among brain regions (Appendix A), which was as expected due to their functional variation [34]. Relative differences in region-specific gene expression for control and treatment groups are depicted as heat maps (where green is increased and red is decreased gene expression; Appendix A) and volcano plots (Figure 1). The degree of change for five genes that showed a significant change in expression due to treatment is detailed in Table 1.

No change in gene expression was detected with the Lipid 70 diet treatment alone in any brain region compared with the control animals. Probiotic HN001 treatment resulted in a 2-fold increase in *Grm4* (metabotropic glutamate receptor) expression in the amygdala (*p* < 0.05) (Figure 1a). The combined lipid and probiotic treatment resulted in reduced expression of GABA-pathway-related genes in the amygdala; a 77% decrease in *Gabre* (GABA_A_ receptor epsilon subunit), a 40% reduction in *Gat3* (GABA transporter), and a 27% decrease in *Gabrg1* (GABA_A_ receptor gamma 1 subunit) (Figure 1b). *Gabrd* (GABA_A_ receptor delta subunit) expression was decreased by 54% in the hippocampus (*p* < 0.05) (Figure 1c). Another 12 genes trended toward altered expression in the amygdala for this treatment group (Figure 1b and Appendix A).

### 2.4. Microbiota

Overall, microbiota profiles were similar between groups (Figure 2a,b). However, some significant differences (FDR < 0.05) in relative abundance between rats fed the control diet and the diet supplemented with Lipid 70 were observed (Table 2). Differences included *Oscillibacter*, unclassified *Clostridiaceae*, *Enterocloster*, and *Faecalibaculum*, which were all more abundant in Lipid 70-fed rats, and *Alistipes*, *Lactococcus*, and *Klebsiella*, which were more abundant in the control diet-fed rats (Table 2). However, only one taxon, unclassified *Enterobacteriaceae*, was significantly altered (FDR = 0.002) by the administration of HN001 (Figure 2c). No significant interactions between diet (AIN control diet and Lipid 70) and probiotic (water and *L. rhamnosus* HN001) were detected for any taxa.

Similar to the differences observed in the caecal microbiome taxonomic composition, numerous functional genes in the caecal metagenome, annotated to the SEED level 4 functional categories, were differentially abundant (FDR < 0.05; |logFC| > 0.263) between rats fed the control AIN diet and the Lipid 70 diet (Appendix A). These genes covered a wide range of functions but in broad terms they generally involved cell envelope or cell capsule genes, energy metabolism, amino acid metabolism and carbohydrate metabolism. Three categories related to antibiotic resistance were also differentially abundant; two were less abundant, while one was higher, in rats fed the Lipid 70 diet compared with rats fed the control AIN diet (Appendix A).

Whereas feeding Lipid 70 led to many changes in functional genes in the caecal metagenome, administration of *L. rhamnosus* HN001 altered the relative abundance of 12 SEED level 4 functions (Table 3). These genes included those related to the cell envelope, amino acid and carbohydrate metabolism, iron acquisition and (microbial) stress response. In this instance, all functions that were significantly different in abundance were more abundant in rats given *L. rhamnosus* HN001 compared with control rats.

As with the comparison of taxonomic composition, no significant interactions between *L. rhamnosus* HN001 and Lipid 70 were detected for gene functions in the caecal metagenomes. The noticeable impact of Lipid 70, but minor impact of *L. rhamnosus* HN001, on the caecal metagenome is illustrated in Appendix A, which shows the hierarchical clustering of SEED level 4 function profiles in the caecal metagenome for functions that exhibit the greatest variation (top 5% coefficient of variation) in abundance across all rats, regardless of group.

### 2.5. Short-Chain Fatty Acids (SCFA)

The total amount of caecal SCFA was not altered in the three individual treatment groups compared with the control (ANOVA) (Figure 3a). The Lipid 70 diet did, however, alter the overall SCFA profile compared with the non-Lipid 70 (control AIN diet)-treated animals (PERMANOVA, *p* = 0.025).

### 2.6. Diet and Brain Lipids

#### 2.6.1. Dietary Lipids

Lipidomics of the dietary lipid composition for the control diet and Lipid 70-enriched diet detected 611 lipid species, with the major species including 41 ceramides, 101 diacylglycerides, 22 hexosylceramides, 59 phosphocholines, 37 phosphoethanolamines, 44 sphingomyelins, and 269 triacylglycerides. The probiotic ingredient was not measured using lipidomics. Of the 611 lipid species measured in the two diets, 530 of these significantly differed (FDR > 0.05) between the two diets, with 380 of these lipids also differing by a 2-fold change as well as in their FDR *p*-value (Appendix A). Of these, 315 were higher in relative abundance for the Lipid 70 diet while 65 were higher in relative abundance for the control diet. Those higher on the Lipid 70 diet included 56 phosphocholines (95%), 19 hexosylceramides (86%), 36 phosphoethanolamines (97%), 43 sphingomyelins (98%), 82 triacylglycerides (30%), 24 ceramides (59%) and 44 diacylglycerides (44%). While the majority of the lipid species detected in the diets were triacylglycerides and diacylglycerides, only 29 triacylglycerides (11%) and 23 diacylglycerides (23%) were increased in relative abundance in the Lipid 70 diet over the control diet.

#### 2.6.2. Brain Lipids

Lipidomics on brain tissue samples (across all five tissue types) detected 443 lipid species. The major classes and numbers of lipids detected included 48 ceramides, 38 diacylglycerides, 51 hexosylceramides, 120 phosphocholines, 62 phosphoethanolamines, 21 sphingomyelins and 62 triacylglycerides. Significant differences were observed between the abundances of all 443 lipid species detected across the five brain regions (ANOVA, FDR *p*-value < 0.05), and this tissue regional variation can be observed in the PCA score plot (Appendix A). However, no treatment effect was detected in the brain region-specific lipid profile for any of the dietary treatments, i.e., for all four treatment groups, for diet alone (control vs. Lipid 70) or for probiotic alone (none vs. probiotic added), with the exception of five lipids in the amygdala region, which showed a probiotic effect. Note, however, that there appears to be no common pattern with these lipids, with one ceramide, one phosphocholine, one phosphoethanolamine and two sphingomyelins, all with differing fatty acids attached.

### 2.7. Comparative Analysis 

Although no significant differences in behaviour were observed as a result of the different treatments, there remains the potential to gain useful insight into the gut–brain relationship by exploring correlations between behaviour and gene expression in the brain that occur with stress, regardless of treatment. Moderate positive and negative correlations were found between behaviour and expression of some genes in the amygdala. These included a range of glutamate- and GABA-receptor and transport-related genes which correlated with the frequency and amount of time spent in different zones of the OFT (Figure 4). Moderate correlations between behavioural test variables and gene expression were also observed in the hippocampus (Appendix A), hypothalamus (Appendix A) and PFC regions (Appendix A). No correlations above a |0.5| threshold were found between the measured behaviour outcomes and the lipid metabolites described in the five regions of the brain, nor were any correlations >|0.5| found between behaviour and the caecal microbiota at the genus level.

## 3. Discussion

The initial finding from this study was that the dietary ingredient treatments did not translate to any detectable changes in behaviour, as evaluated in three tests of stress, anxiety and memory. Some of the diet treatments did, however, result in specific changes in gene expression in key brain regions and produced changes in the caecal microbiota, but these did not alter the brain lipidome. The ability of *L. rhamnosus* HN001 to reduce anxiety and depression in humans [16], but not in the WKY animal model, suggests that this depressive phenotype was too severe for the food intervention to alter behaviour. *L. rhamnosus* HN001 did, however, affect the expression of genes in pathways associated with fear, anxiety and depression in the amygdala (*Grm4*) and produced minor changes in the microbiota, implicating actions on GBA pathways in the WKY strain. We note that *L. rhamnosus* HN001 given over six weeks reduced stress-related behaviours in the OFT and EPM tests, in which stress was induced through an unpredictable chronic mild stress paradigm in Sprague Dawley rats [35].

Contrary to expectation, the Lipid 70 diet did not alter gene expression or lipid composition in any brain region but instead had effects that were exclusive to the large intestine (caecum), producing shifts in the caecal microbiota and SCFA profiles indicative of broad effects on digestion, caecal fermentation and metabolism. A new finding in this study was the demonstration that the *L. rhamnosus* HN001 + Lipid 70 ingredients acted synergistically to affect gene expression in specific functional neural pathways, reducing GABA_A_ subunit expression in the amygdala and hippocampus. 

### 3.1. L. rhamnosus HN001 Induces Specific Change in Brain Gene Expression and Caecal Microbiota

The increased expression of *Grm4* in the amygdala is consistent with *L. rhamnosus* HN001 altering GABA–glutamate neurotransmitter-receptor pathways in brain regions associated with anxiety. *Grm4* is functionally important in neurons because it is located pre-synaptically, where it modulates the general release of neurotransmitters, including those for glutamatergic, dopaminergic, GABAergic and serotonergic signalling throughout the brain, and post-synaptically, by modulating the neural effects of glutamate [36]. In humans, the amygdala links pain sensation with negative emotions, and activation of pathways associated with *Grm4* has been found to reduce anxiety and depression where this is a symptom of persistent pain [37]. Both vagal and dorsal root ganglia in the enteric nervous system (ENS) are capable of activation by glutamate signalling to key brain regions [38]. Live *L. rhamnosus* HN001 is able to reach the caecum [39]; thus, a direct effect of this on the ENS is considered possible. The probiotic *L. rhamnosus* JB-1 has been shown to alter the expression of GABA_A_ receptors in key brain regions in mice, and it also reduced anxiety and depressive behaviours (including in the EPM) and stress-induced corticosterone levels in a manner that required vagus signalling [40]. 

Although the only taxon that was elevated in the caecum by feeding *L. rhamnosus* HN001 was the *Enterobacteriaceae* family, a relatively abundant range of gene functions were altered. These included genes involved in the microbial stress response and iron acquisition. *Enterobacteriaceae* and other facultative anaerobes possess numerous mechanisms for scavenging iron, and their metabolism of iron can influence health and disease [41]. Interestingly, iron status has been consistently shown to decrease proportions of *Lactobacillaceae* [42], which points to a potential mechanism of interaction between *L. rhamnosus* HN001 and *Enterobacteriaceae*. At the same time, the process of iron metabolism by *Enterobacteriaceae* has been shown to increase the physiological stress response of other members of the microbiome [43]. That such a specific change in the caecal microbiota by *L. rhamnosus* HN001, albeit small, was associated with the increased expression of a single gene (*Grm4*) in the amygdala might suggest that a particular physiological change affecting a gut-to-brain signalling modality is possibly involved. Indeed, there is evidence that simultaneous iron dysregulation and microbial dysbiosis can exacerbate Alzheimer’s type dementia [44], although the mechanisms behind this remain unclear. Furthermore, maternal iron deficiency in a mouse model can lead to offspring exhibiting decreased BDNF expression in the brain, spatial learning deficits and an altered faecal microbiota profile [45].

### 3.2. MFGM Treatment Alters the Caecal Microbiota, SCFA

The measured effects of the Lipid 70 diet were substantially different from that of the diet supplemented with *L. rhamnosus* HN001. Treatment with Lipid 70 had no effect on brain gene expression and produced a much more significant shift in the caecal microbiota profile. The changes caused by Lipid 70 treatment, including increases in some *Clostridiaceae*, *Lachnospiraceae*, and *Ruminococcaceae*, suggest an increase in fermentative capacity and SCFA production, as supported by the SCFA analysis (butyric acid *p* = 0.053; valeric acid, acetic acid, succinic acid, *p* = 0.052; propionic acid; *p* = 0.071). This premise was further supported by the analysis of functional genes where numerous carbohydrate-related SEED level 4 functions were significantly more abundant in Lipid 70-fed rats. The Lipid 70-induced changes in the caecal SCFA content does hint at potential benefits to brain function. For example, butyrate is a known inhibitor of histone deacetylases (HDACs), and suppression of HDACs has been shown to stimulate neurogenesis [46] and enhance BDNF expression and memory [47]. However, any microbiome/butyrate-mediated beneficial impact of Lipid 70 on brain function would need to be confirmed experimentally. Furthermore, it is possible that the increased SCFA content was simply derived from a higher dietary SCFA content found in the Lipid 70 ingredient.

This lack of effect might be age-related or due to the exacerbated stress of having been separated out into single caging four weeks prior, particularly for the WKY stress-prone strain. We note that when provided to growing rats, MFGM improves spatial memory and alters the gut microbiota (postnatal day 21–56) [20] and also increases neuromuscular development (postnatal day 10–69) by altering the muscle fibre-type profile [48]. However, the inability of MFGM to alter recognition memory in the NOR test in younger (2 mo old) rats [20] is consistent with our results for Lipid 70 in older (6 mo old) rats. When given together with prebiotics and lactoferrin, MFGM improves sleep quality under stress (postnatal day 24–94) and alters the gut microbiota [49].

The lack of change in the brain lipid profile in response to the Lipid 70 diet was contrary to prediction and suggests a lack of sufficient nutritional lipid uptake to alter brain tissue composition in this rat model. When fed to growing pigs (10–34 day old), a related lipid product (Lipid 100) produced changes in brain lipid composition, particularly in the hippocampus [50]. Irrespective of species, the difference in results between studies may have been due to the piglet study being carried out during a time of rapid animal growth, enabling dietary lipids to be incorporated into the brain during this phase. Alternatively, the relative immature state of the GI tract in the piglets, compared with the adult rats, may have led to greater absorption of the lipids into the body.

### 3.3. MFGM Lipid 70 and L. rhamnosus HN001 Together Alter Expression in GABA-Mediated Pathways

The most pronounced effect on gene expression in the brain was detected when Lipid 70 was given in combination with *L. rhamnosus* HN001. That the gene expression changes detected were distinctive from that for either treatment alone exemplifies a synergistic effect. The decreased expression of genes for three GABA_A_ receptor subunits (*Gabre* and *Gabrg1* in the amygdala and *Gabrd* in the hippocampus) and a GABA transporter (*Gat3* in the amygdala) denotes a specific effect of *L. rhamnosus* HN001 + Lipid 70 treatment on GABA activation pathways and in specific tissues important for stress, anxiety, learning and memory. That *L. rhamnosus* HN001 + Lipid 70 treatment altered brain neurotransmitter/receptor pathway expression in the amygdala is of note because this region is associated with anxiety vulnerability in WKY rats during stress [51].

Decreased expression of genes was observed with *L. rhamnosus* HN001 + Lipid 70 treatment. Those that approached significance included the α2, β1, γ1, and ε GABA_A_ receptor subunits that are distinct in being rare and expressed at high levels in the amygdala, where they are considered likely to assemble into pentamers in rats [52] and humans [34]. A reduction in the expression of these receptors would presumably reduce their inhibitory influence in the amygdala. The probiotic *L. rhamnosus* JB-1 has been reported to reduce expression of the GABA_A_ α2 receptor subunit in the mouse amygdala, with a corresponding decrease in anxiety and depression-related behaviours [40]. Although we did not detect measurable behavioural changes in stress/anxiety or memory in response to *L. rhamnosus* HN001+ Lipid 70, the reduced expression of genes coding for subunit components of GABA_A_ receptors that, when activated, dampen neuronal excitability suggests that a neurochemical shift toward reduced inhibition, and thus increased excitability, may have been occurring in the amygdala. We note that *L. rhamnosus* JB-1 was not effective in vagotomised mice [40]. Since WKY rats are considered to have increased vagal tone, this may have masked any vagally communicated signals from *L. rhamnosus* HN001.

*Gat3* is a high affinity GABA plasma membrane transporter that is expressed solely in astrocytes in rodents rather than at the synapse in the cerebral cortex and hippocampus [53,54]. In the rat hippocampus, *Gat3* regulates levels of extrasynaptic GABA, mopping up excess GABA (released in response to action potentials and independent release) by transporting it into astrocytes [54]. Thus, *Gat3* plays a functionally important role in modulating the level of tonic activity by GABA via its availability. If it has a similar role in the amygdala, its reduced expression in response to the *L. rhamnosus* HN001+ Lipid 70 treatment would be expected to slow the removal of GABA from the synaptic cleft, potentially affecting GABA involvement (strengthening inhibition) in fear/anxiety pathways.

The decreased expression of the GABA_A_ receptor delta subunit in the hippocampus in *L. rhamnosus* HN001+ Lipid 70-treated animals is of functional significance because δ subunit density is highly expressed in dentate gyrus granule cells, and the δ subunit-containing GABA_A_ receptors are assumed to comprise the majority of extrasynaptic receptors that mediate tonic inhibition in the amygdala. This is functionally important in the control of cellular excitability because αβδ-containing GABA_A_ receptors are highly sensitive to GABA with minimal desensitization and are considered a key mediator of tonic inhibition in the central nervous system [55]. A decrease in the level of expression of delta-containing GABA_A_ receptors (as conferred by *L. rhamnosus* HN001+ Lipid 70 treatment) extrasynaptically might reduce the inhibitory basal tone they normally confer, leading to increased cellular excitability. Thus, the treatment decreased the expression of a gene that contributes to an overarching inhibitory mechanism with the potential to increase excitatory pathways and counter anxiety and depression.

Other genes that trended toward decreased expression predominantly belonged to the GABA/glutamate pathway, as well as to serotonergic and nitrogenergic pathways. Furthermore, the fact that the changes were specific to amygdala and hippocampus regions and were not detected in cerebellum points to a functional specificity for the combined dietary treatment. The treatments also did not result in any alteration in the selected lipid metabolism or immune-related (cytokine) genes. This is of note because immune pathways have been reported be altered in the amygdala during stress [56,57].

### 3.4. Gene Expression and Behaviour Correlations

Correlations with gene expression in four regions of the brain (hypothalamus, amygdala, hippocampus, prefrontal cortex) were only found in the OFT and EPM tests. Of those measurements, only time spent (s) in the different arena zones and animal movement, either in time (s) or distance (cm) during the test, showed moderate correlations. Within these correlations, gene expression, predominantly in the hypothalamus, showed contrasting associations with the time spent in the more exposed zones versus the sheltered zones of the OFT. This pattern was also present, to a lesser extent, for gene expression in the amygdala. While the dietary treatments from this study were not a factor in these associations, we were able to demonstrate links between gene expression and the measurements used to describe anxiety and activity-like behaviours. We note an overlap between changes in gene expression in the amygdala with diet treatment and those that correlate with activity in the OFT for *Grm4*, *Gabre* and *Gat3*, and to a lesser extent for *Gabra2*, *Grik1*, *Grik2* and *Nos1*. This seems to highlight the importance of GABA_A_ receptors and transport in determining behaviour. Together with our finding that the ingredients altered GABA gene expression, this suggests the potential for diet interventions to alter behaviour by targeting GABA pathways.

### 3.5. Limitations

The lack of any detectable effect of the dietary treatments on measures of anxiety and depression-related behaviours was surprising. We acknowledge some potential confounders in the behavioural tests. First, WKY is a selectively bred anxious rat strain known to display inhibited behaviour characterized by long periods of immobility [58] and fewer movements between segments in the open-field test [59]. Regardless of treatment, all rats in our study demonstrated high levels of thigmotaxis in the open field, which is an indicator of anxiety [60]. Therefore, it is possible that the treatments were not influential enough to override the strong anxiety of this strain. Furthermore, side preference may have confounded our NOR results; we acknowledge that placement of the novel object should be randomized [61]. Rat age is another possible factor; at four months of age, the brain would be considered less neuroplastic than at weaning in terms of synapse structure and functional connectivity involved in memory and learning [62]. Finally, the rats in our study were first housed in groups and then switched to individual cages prior to the treatments beginning. Single housing can cause increased anxiety and depression in some rat strains [63] and reduce feed intake and resilience [64] to challenges (e.g., [65,66]).

## 4. Materials and Methods

### 4.1. Ethical Approval

This study (AE14714) was approved by the AgResearch Grasslands Animal Ethics Committee (Palmerston North, New Zealand) according to the New Zealand Animal Welfare Act (1999). 

### 4.2. Animals

Sixty-three male Wistar Kyoto (WKY) rats were imported from the Animal Resources Centre (Canning Vale, WA, Australia) at 9.6 ± 0.3 weeks of age. The animals were allowed to age to adulthood in the facility over 3.5 months in group housing at a constant temperature of 19 ± 1°C and maintained under a 12/12 h light/dark schedule (lights on at 7:00 a.m.). During this time, the rats were provided with LabDiet—Prolab RMH 1800 and water *ad libitum*. They were handled several times weekly over the 3.5 months and then more regularly in the month prior to the behavioural testing. One week prior to the commencement of dietary treatments, the animals were housed in pairs. One day prior to the dietary treatments starting, they were housed individually. At 23.6 ± 0.3 weeks of age, the animals were assigned in blocks to one of three treatment groups (n = 15–16 per group) or a control group, distributing evenly across weight (383 ± 5 g) and prior housing groups. Rats were fed an adult maintenance diet (AIN-93M; OpenStandard Rodent Diet, Research Diets, Inc. New Brunswick, NJ, USA) and water, provided ad libitum. Animals were monitored daily for liquid intake, every 3 days for food intake and weekly for weight and to measure their General Health Score (1–5; NZ Animal Health Care Standard). The day following completion of the behavioural tests, the animals were euthanised using carbon dioxide inhalation overdose.

### 4.3. Dietary Treatments

The dietary ingredients were provided by the Fonterra Research and Development Centre, Palmerston North, New Zealand.

#### 4.3.1. Milk Fat Globule Membrane and Whey Dietary Supplementation

A dairy whey extract enriched in milk fat globule membrane (MFGM), SurestartTM MFGM Lipid 70 (Fonterra Co-Operative Group, Auckland, New Zealand), was used. The macronutrient composition of Lipid 70 was 72% protein, 20% total fat, 5% phospholipid and 3% lactose.

The rat diets were composed of the OpenStandard Modified Rodent Diet (Research Diets, Inc. New Brunswick, NJ, USA) (kcal %): 12% fat, 13% protein, 75% carbohydrate (cornstarch, maltodextrin, sucrose), cellulose, BW200, L-cystine, choline bitartrate, minerals and vitamins. The diets were nutritionally balanced for macronutrient content. The Lipid 70 treatment diet contained 8% Lipid 70. Based on an anticipated intake of 20 g per day, this was calculated at a dose of 1.6 g of Lipid 70 per day. In order to maintain a constant percentage of fat, the soyabean oil in the standard diet was reduced to account for the milk fat from the dairy extract. The control diet protein component contained 100% casein, whereas the test treatment diet was 54% casein and 46% whey. 

#### 4.3.2. Probiotic Supplementation 

The probiotic used was *Lacticaseibacillus rhamnosus* strain HN001 (LactoB HN001™). *L. rhamnosus* HN001 was added to the water and supplied to the animals ad libitum. Based on their anticipated average daily intake, this would provide a dose of approximately 10^9^ CFU/day.

#### 4.3.3. Treatments

Control animals were provided an adult maintenance solid diet of AIN-93M ad libitum, which contained casein as the protein source, and water. Treatment groups were provided with either (1) the Lipid 70 diet and water, (2) AIN-93M and *L. rhamnosus* HN001 in water, or (3) both the Lipid 70 diet and *L. rhamnosus* HN001 in water.

### 4.4. Study Design

Power analysis at 80% based on anticipated behavioural differences indicated that 16 animals per group were required. The animals were assigned to one of four groups: control diet + water, Lipid 70 diet + water, control diet + *L. rhamnosus* HN001, Lipid 70 diet + *L. rhamnosus* HN001. Animals were fed their respective diets for 34 days and assessed in the novel object recognition (NOR) test on days 29 to 31, the open field test (OFT) on day 32, and the elevated plus maze (EPM) test on day 33. Rats were then euthanised on day 34 and biological samples collected. The experiments were carried out in three blocks.

### 4.5. Behavioural Testing 

The animals’ behaviour was video-recorded during the light phase of the light/dark cycle (between 8:00 a.m. and 12:00 p.m.). A single camera (HERO 7, GOPRO, San Mateo, CA, USA) was positioned 130 cm above the testing arena, which was set up in a room adjacent to the housing room. Rats were placed in the testing arena each day in the same order and the surfaces and objects cleaned with 70% ethanol between animals.

Open field test (OFT). The OFT (opaque arena: 90 cm diameter, 50 cm high) (day 32) was performed as per [67] for 10 min (lux: 900 at arena centre) to capture behaviours indicative of anxiety and exploration. 

Elevated plus maze (EPM). The EPM (Panlab, Harvard Apparatus) (day 33) was performed for 5 min (lux: 200 for open arms, 60 to 90 for closed arms) to capture behaviours indicative of anxiety.

Novel object recognition (NOR). This test was used to test recognition memory and was carried out over three consecutive days. Lighting for the NOR (opaque rectangular arena 42 cm × 60 cm × 38 cm) was 60 lux in the centre of the area. First, the rats underwent habituation (day 29), where they were allowed to explore the empty arena for 10 min [68]. Then, on the pre-test day (day 30), they were introduced to two identical objects (circular opaque grey tube: 15 cm high × 7 cm diameter, 0.5 cm thick, with an open top) for 5 min. On the main test day (day 31), the object on the left side was switched for a different one (square-shaped opaque white tube: 15 cm high, 7 × 7 cm square, 0.5 cm thick, with an open top) for 5 min.

The video was analysed using EthoVision XT 10 (Noldus, Wageningen, The Netherlands). Variables for the OFT and EPM test were recorded using the centre-point of the rat as the reference point. For the NOR, the nose-point was used for the analysis of exploration of the objects. The experimenter conducting the analysis was blinded to the treatment groups. To ensure reliability of our analysis, six videos from each test were randomly selected and re-analysed on EthoVision; this resulted in 9.5% of the observations being used to calculate the intraclass correlation coefficient (mean ICC: 0.999; range: 0.998–1.0). EthoVision automatically recorded the following variables: mean velocity (cm/s), max velocity (cm/s), total distance moved during the test (cm), total frequency of movements during the test (no.), as well as the duration (s) and frequency (no.) spent in each of the zones of interest (OFT: 5 concentric zones; EPM: open arms, closed arms, centre point). For NOR, using the day 3 results only, the duration of time spent with the novel object versus the familiar object was used to calculate the relative duration.

### 4.6. Sample Collection

Animal samples (brain tissue, caecal content and plasma) were collected immediately after euthanasia. Brain tissue samples (10–20 mg) were dissected from the hippocampus, amygdala, hypothalamus, cerebellum and prefrontal cortex regions. Half of each region was stored in RNAlater (Ambion, Life Technologies, Carlsbad, CA, USA) for 24 h then transferred to −20 °C for gene expression analysis, and the other half was snap-frozen in liquid nitrogen and stored at −80 °C for lipid analysis. Post-mortem blood samples were drawn by cardiac puncture using a 21-gauge needle and syringe, pre-rinsed with EDTA, into lithium heparinised vacutubes, centrifuged at 2000× *g* for 5 min at room temperature, and the plasma collected and frozen at −80 °C. Caecal contents were snap-frozen in liquid nitrogen and stored at −80 °C.

### 4.7. Gene Expression

#### 4.7.1. RNA Isolation

RNA was isolated from brain tissue using the RNeasy Lipid Tissue Mini kit (Cat. No. QIAG74804, Qiagen, Valencia, CA, USA) and QIAzol Lysis Reagent (Qiagen, Valencia, CA, USA) according to manufacturer’s instructions. RNA quantity was assessed using a Bioanalyser (Bio-analyzer 2100, Agilent Technologies, Palo Alto, CA, USA).

#### 4.7.2. RNA Analysis Using Nanostring

Gene expression analysis was carried out using the nCounter Analysis System (NanoString Technologies Inc., Seattle, WA, USA) in conjunction with standard Elements chemistry protocols for a custom CodeSet. Custom-designed rat gene-specific oligonucleotide reporter and capture probe pairs (Integrated DNA Technologies, PTE LTD, Singapore) and nCounter Elements™ TagSets (NanoString Technologies) were used. The CodeSet consisted of probes for 10 reference genes (5HTT/Slc6a4, Abcf1, Gusb, Hprt1, Ldha, Rplp0, Polr1b, Snap25, Stx1a, Vglut1/Slc17a7) and probes specific for a further 86 genes (Appendix A) selected based on their known or implicated involvement in receptor neurotransmission pathways associated with anxiety, depression and memory. These included genes for BDNF, corticotropin-releasing hormone, cytokine receptors, lipid metabolism enzymes, subunits for GABA, glutamate, serotonin, dopamine, purinergic and nitrogenergic receptors, and associated proteins and enzymes for the synthesis and degradation of associated neurotransmitters (Appendix A).

### 4.8. Caecal Microbiota

DNA extraction and shotgun metagenomic sequencing was carried out as previously described, with the exception that host reads were identified and removed by aligning against the rat genome (Rnor_6.0 release 102) as reference. DIAMOND version 0.9.22 [69] was used to map reads against the “nr” NCBI database. Megan version 6 ultimate edition [70] was used to assign taxonomy and putative functions to the DIAMOND alignment files against the SEED Subsystems database [71]. Differences in taxonomic composition were analysed using ANCOM-BC [72], and differences between functional gene abundances were determined using likelihood ratio tests in edgeR [73]. The *p*-values were adjusted for multiple testing using the false-discovery rate (FDR) method, with FDR < 0.05 considered significant.

### 4.9. Short-Chain Fatty Acid (SCFA) Analysis

SCFA analysis was carried out using 3-nitrophenylhydrazine (3NPH) derivatisation and liquid chromatography–mass spectrometry (LC-MS) quantified with stable isotope standards, according to methods adapted from the methods of Han et al. (2015) [74] and Leibisch et al. (2019) [75]. Briefly, 10–30 mg of wet caecal contents was added to 1 mL 70% isopropyl alcohol and the mixture homogenised, after which, 300 µL of sample homogenate was dried overnight on a speed vacuum concentrator to calculate the dry weight. Aliquots of the dried homogenates were placed in 2 mL glass autosampler vials to an equivalent dry matter content of 2 mg dry weight/mL, to which 50 µL water, 20 µL of 200 mM 3NPH solution and 20 µL of 120 mM EDC/pyridine solution were added. Vials were then heated at 40 °C for 30 min; then, 200 µL water and 50 µL of stable isotope internal standard mixture were added prior to analysis. 

LC-MS analysis was performed on a Thermo Q-Exactive LC-MS system (Thermo, Waltham, MA, USA) using an atmospheric chemical ionisation probe in negative ionisation mode. Chromatographic separations were performed on a Thermo Hypersil GOLD UHPLC column (100 × 2.1 mm, 1.9 µm particle size) using gradient elution with 95% water: 5% acetonitrile and 100% acetonitrile as mobile phases A and B, respectively. The column flow rate was 350 µL/min and the column temperature 40 °C. The binary solvent elution gradient was optimized at 15% B for 2 min, 15–55% B for 9 min and then held at 100% B for 1 min. The column was equilibrated for 3 min at 15% B between injections. Data for each SCFA were calculated as µmole acid/g caecal content per animal to enable comparison.

### 4.10. Brain Lipid Metabolomics

Lipid extractions from a single aliquot of brain tissue were performed using a biphasic extraction (modified Folsch extraction) as previously described [76,77]. Brain extracts were analysed on a Shimadzu LC-MS 9030 Q-Tof system using the gradient and solvent system previously described [76]. LCMS data files were converted to the centroid mzML format using the Shimadzu file converter and then explored using the open-access software package MS-DIAL ver 4.8 [78].

### 4.11. Statistical Analysis

#### 4.11.1. Animal Metrics

Weight gain in animals was measured as the percentage change in weight in contrast to the weight at pre-treatment. The impact of the percentage change in weight was investigated using a mixed-effects model, with the fixed effects including time as a covariate, pre-study animal weight, whether an animal was given liquids that contained probiotic or not (liquid) and whether an animal’s diet contained lipids or not (food) as well as the interaction between liquid and food. 

A mixed-effects model was fitted with food intake converted into the difference in intake on subsequent days as the outcome. The fixed effects were time as a covariate, whether the animal was given probiotics or not (liquid), whether the animal was fed probiotics in their diet or not (food) and the interaction between liquid and food; the random effect was rat. 

#### 4.11.2. Animal Behaviour Tests

Elevated plus maze (EPM). A range of behaviours were measured in the EPM test as response variables: mean velocity, maximum velocity, distance moved, movement frequency, movement duration, closed-arm movement duration, centre movement duration, closed-arm movement frequency and centre movement frequency. One variable, duration spent in the open arms, was discarded as the animals collectively spent so little time in that space that there was very little variation in the data. In addition, the data from two rats (Lipid 70 + *L. rhamnosus* HN001 group) were excluded as the EthoVision detection was not successful. With the exception of two variables (closed-arm duration and centre duration), the response variables were investigated using an analysis of variance (ANOVA) with the weight of an animal pre-trial, their probiotic and lipid intake as well as the interaction between the latter two factors as explanatory variables. The durations spent in the closed-arm space and centre space were converted into a percentage of the total monitored duration (to correct for EthoVision tracking errors). These percentages were analysed using beta-regression models with the same predictor variables as the ANOVA models. 

Open field test (OFT). The same range of behaviours were measured as in the EPM test (with the exception of the open and closed arms being switched out for the five concentric zones); for consistency, we fitted the same models. The behaviours were analysed concurrently as a multivariate response using a permutational multivariate analysis of variance (PERMANOVA), with the treatment groups and pre-treatment weight as predictor variables.

Novel object recognition (NOR). The outcome variable of interest was the relative duration, that is, the difference in time spent with the novel object (left) minus the time spent with the familiar object (right) divided by the total time for both objects. Positive values indicate more time spent with the novel object. This response was analysed with pre-trial weight, liquid, food, the interaction between liquid and food as well as an indicator function for whether an animal spent more time with the left object on days 1 and 2.

#### 4.11.3. Short-Chain Fatty Acid Analysis

Differences in SCFA profiles were compared using PERMANOVA, with the fatty acid measurements as the response variables and the four treatment groups as the explanatory variables.

#### 4.11.4. Gene Expression

Analysis was performed using Nanostring nCounter nSolver™ 4.0 (Nanostring MAN-C0019-08) with the Nanostring Advanced Analysis Module 2.0 plugin (Nanostring MAN-10030-03) while following the Nanostring Gene Expression Data Analysis Guidelines (Nanostring MAN-C0011-04). The reference genes used for normalisation were *Slc6a4* (5HTT), *Abcf1*, *Gusb*, *Hprt*, *Ldha*, *Rplp0*, *Polr1b*, *Snap25*, *Stx1a* and *Slc17a7* (VGLUT1). Analysis of genes expressed in each of the five brain regions was conducted separately. Genes were considered differentially expressed with a log_2_FC > 0.263 (FC > 1.2) and a Benjamini–Hochberg false-discovery rate (FDR) correction of *p* < 0.05. When “block” was treated as a random event, there was no significant difference detected between blocks.

Functional analysis was conducted on genes (excluding reference genes) grouped by function: GABA (22), glutamate (21), serotonin (12), immune (10), lipid (9), purinergic (2), nitrogenergic (1), catecholamine (adrenergic, noradrenergic, dopaminergic) (5), neuronal general (2), hormonal (1) and stress specific (1). Two-factor non-parametric permutation MANOVA tests were conducted in R (v4.0.2) and the Vegan package. Gene expression within function groups was considered significant at *p* < 0.05.

#### 4.11.5. Brain Lipid Metabolomics

Metabolomics data analysis was performed using MetaboAnalyst v5.0 [79] and the statistical software package SIMCA (v16.0). The peak-area data were log2-transformed and auto-scaled. Univariate and multivariate data analyses were conducted and ANOVA for each brain region carried out. A false-discovery rate (FDR) correction was utilized to reduce the risk of false positives.

#### 4.11.6. Comparative Analysis

Data were compared among measured parameters for behaviour, gene expression, lipid metabolomics and caecal microbiota. Canonical sPLS comparative analyses were performed using the mixOmics package for R (v4.0.2) [80]. Due to missing data, a total of 9 out of 63 rats were excluded from the analyses. These animals were removed from the control (3), *L. rhamnosus* HN001 (3), and Lipid 70 + *L. rhamnosus* HN001 (3) treatment groups. Correlations were made between selected behaviour measurements and the gene expression for each of the five brain regions, between the behaviour dataset and lipid metabolomics for each of the five brain regions, and between the behaviour dataset and caecum microbiota (genus level). A threshold of >0.5 was used to indicate a moderate or greater correlation. NOR test movements and non-movements are described as cumulative(s).

## 5. Conclusions

The results from our study demonstrate that a probiotic (*L. rhamnosus* HN001) with postpartum anxiolytic and anti-depressive properties was able to increase the expression of a key excitation modulatory glutamate receptor in the amygdala. These changes coincided with changes in the microbiome, including elevated caecal *Enterobacteriaceae* and an increased abundance of microbial genes involved in iron acquisition and the microbial stress response. Together, these findings suggest that a specific gut-to-brain association may be occurring. Our findings demonstrate synergistic physiological changes exerted by the *L. rhamnosus* HN001 + Lipid 70 ingredient combination in reducing expression of GABA_A_ receptor genes, suggesting neurochemical changes in key neural pathways involved in fear, anxiety, depression and memory. Although no direct psychophysical changes were detected due to treatment, our behavioural analyses revealed broader correlations with gene expression, reflecting the subtle nature of physiological changes elicited. The data from our study paves the way for a mechanism-driven approach to ingredient selection based on the targeting of specific brain-function pathways.

## Figures and Tables

**Figure 1 ijms-24-10433-f001:**
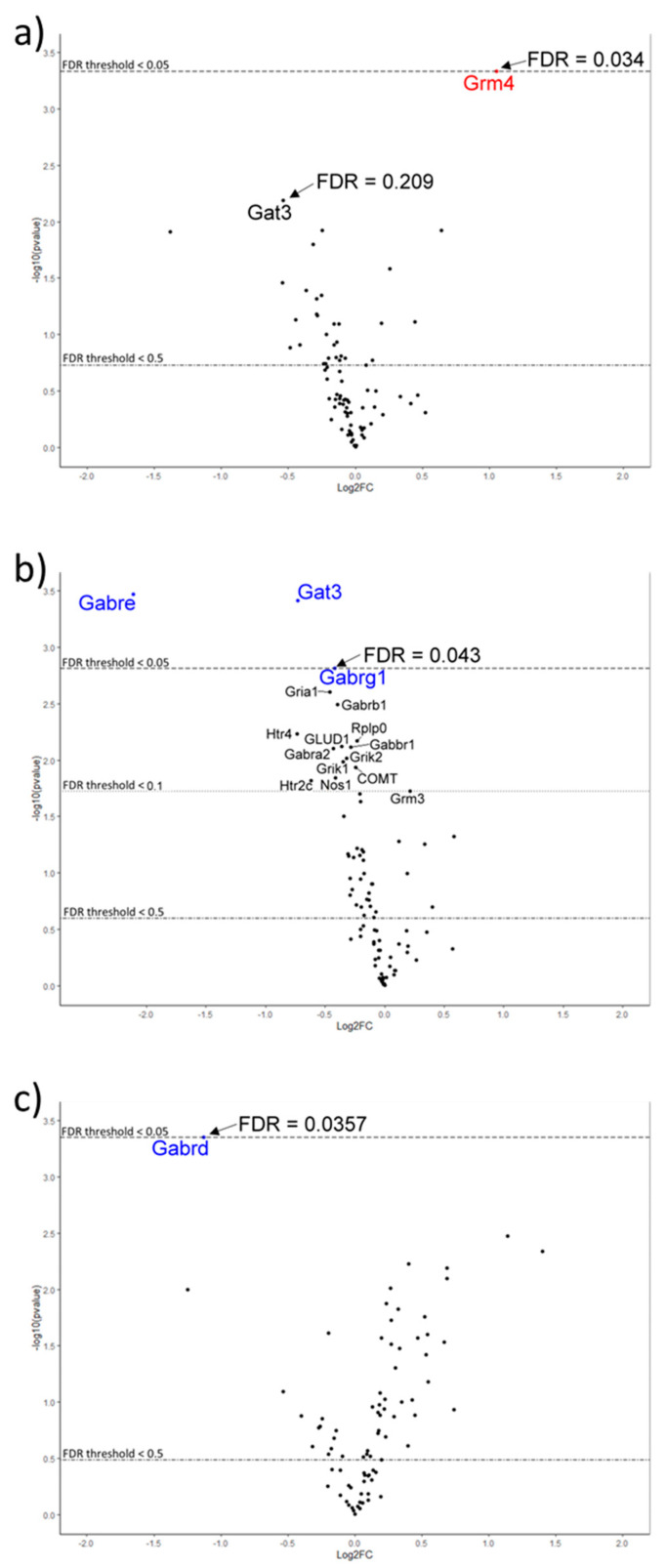
Volcano plots showing differential gene expression (DE) in brain-tissue regions of rats who received various treatments relative to control rats: (**a**) Amygdala: increased DE (red) of the *Grm4* gene in the *Lacticaseibacillus rhamnosus* strain HN001 (LactoB HN001™) treatment group; (**b**) Amygdala: decreased DE (blue) of *Gabre*, *Gat3* and *Gabrg1* genes in the *L. rhamnosus* HN001 + Lipid 70 (Surestart^TM^ MFGM Lipid 70) group; (**c**). Hippocampus: decreased DE (blue) of the *Gabrd* gene in the *L. rhamnosus* HN001 + Lipid 70 group. Thresholds were based on an FDR correction of (*p* < 0.05) and a log_2_fold change > 0.263 (FC < 1.2).

**Figure 2 ijms-24-10433-f002:**
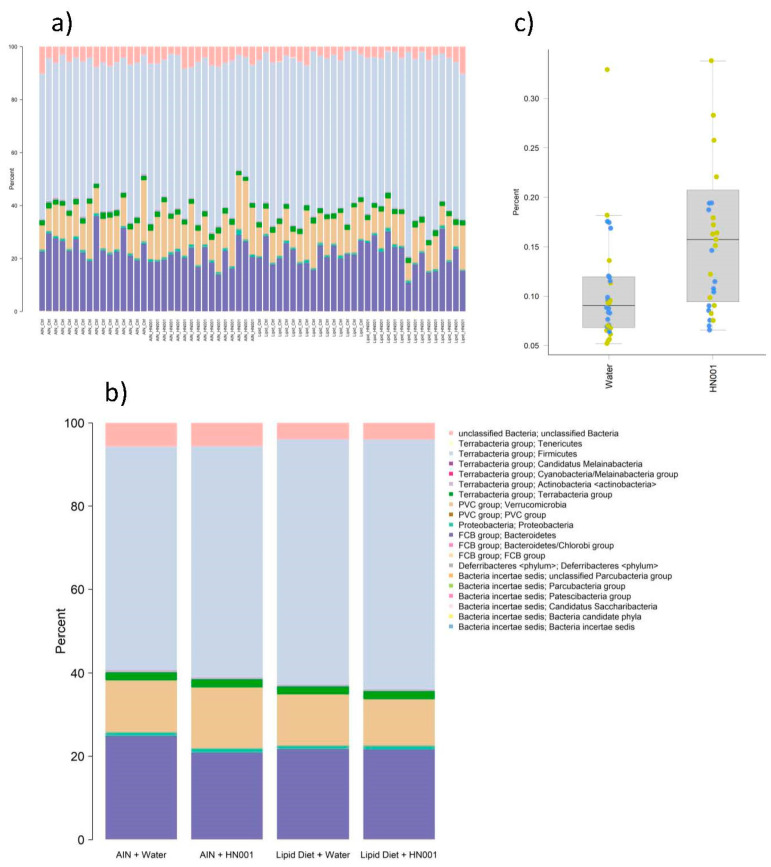
Caecal microbiome. Composition profiles at the phylum level for (**a**) individual rats and (**b**) group means. (**c**) Boxplot of caecal relative abundance of unclassified *Enterobacteriaceae* in rats provided with plain drinking water (non-HN001) or water supplemented with the *Lacticaseibacillus rhamnosus* strain HN001 (LactoB HN001™). Points show individual rats coloured by diet group (non-Lipid 70, yellow; Lipid 70 (Surestart^TM^ MFGM Lipid 70), blue).

**Figure 3 ijms-24-10433-f003:**
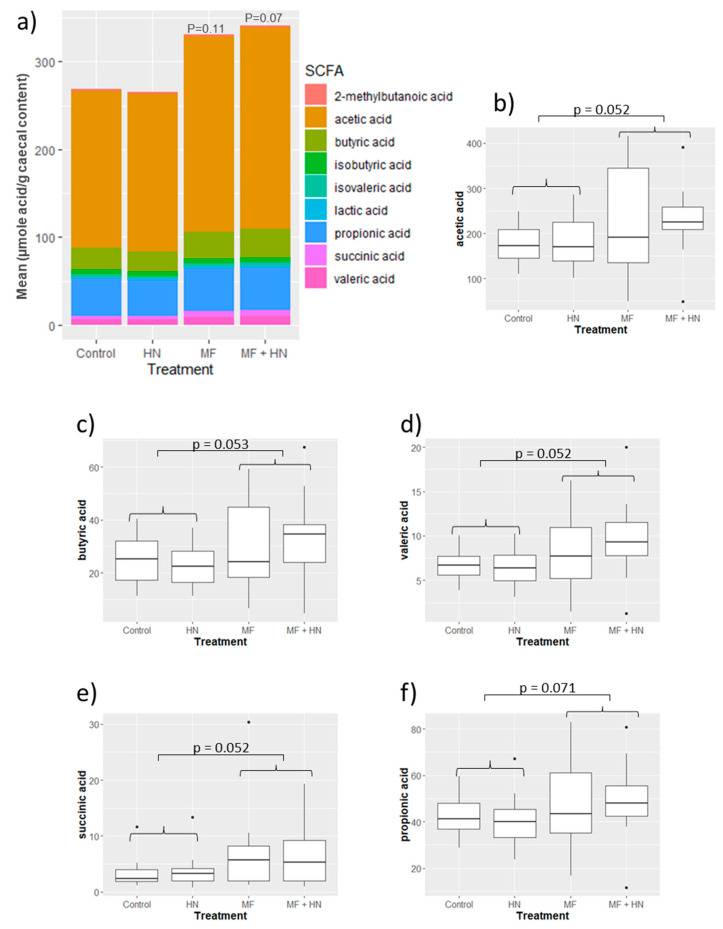
Short-chain fatty acids (SCFA) in caecal content. Differences in SCFA among treatment groups are shown as a bar chart (**a**) comparing overall profiles, and (**b**–**f**) as box plots showing median SCFA levels (μmole acid per g caecal content) with 95% confidence intervals (*p*-values from ANOVA are shown). HN—*Lacticaseibacillus rhamnosus* strain HN001 (LactoB HN001™); MF—milk fat globule membrane (Surestart^TM^ MFGM Lipid 70).

**Figure 4 ijms-24-10433-f004:**
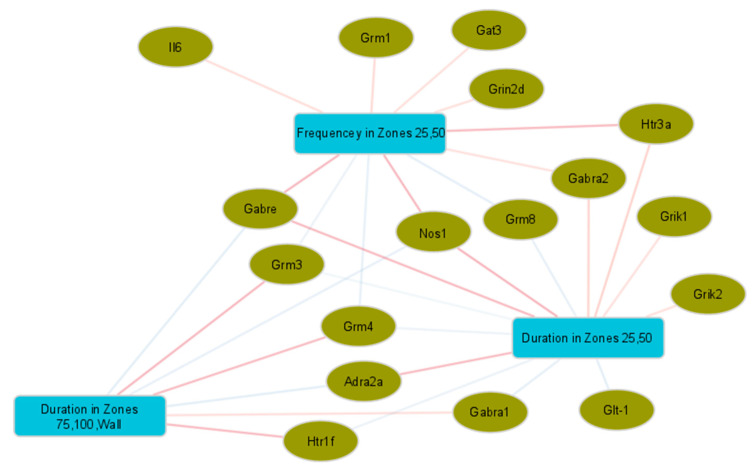
Sparse partial least-squares (sPLS) canonical analysis of open-field behavioural test measurements and gene expression in the amygdala. Network showing variables with canonical correlations > |0.5|, with genes indicated by ellipses and behavioural measurements by rectangles. Edge colours indicate the direction of correlations between variables, with positive values shown in red and negative values in blue.

**Table 1 ijms-24-10433-t001:** Differential brain gene expression analysis using a lm.nb (log-linear negative binomial model) within the nSolver 4.0 software.

Tissue	Treatment	Gene	log2 (FC)	FDR	Expression
amygdala	HN001	*Grm4*	1.05	0.034	increased
amygdala	HN001 + Lipid 70	*Gabre*	−2.11	0.0163	decreased
amygdala	HN001 + Lipid 70	*Gat3*	−0.73	0.0163	decreased
amygdala	HN001 + Lipid 70	*Gabrg1*	−0.418	0.043	decreased
hippocampus	HN001 + Lipid 70	*Gabrd*	−1.13	0.0357	decreased

Significance values and fold changes are relative to the control group (AIN diet and water). The genes that were significantly differentially expressed were *Grm4* (metabotropic glutamate receptor), *Gabre* (GABA_A_ receptor epsilon subunit), *Gat3* (brain-specific GABA transporter), *Gabrg1* (GABA_A_ receptor gamma 1 subunit), *Gabrd* (GABA_A_ receptor delta subunit). HN001—*Lacticaseibacillus rhamnosus* strain HN001 (LactoB HN001™); Lipid 70—milk fat globule membrane (Surestart^TM^ MFGM Lipid 70).

**Table 2 ijms-24-10433-t002:** Taxa with significantly different relative abundances (ANCOM FDR < 0.05) between rats fed the control AIN diet or the diet supplemented with Lipid 70 (Surestart^TM^ MFGM Lipid 70); mean percent ± SEM (n = 16 per treatment group).

Phylum	Family	Genus	Control Mean	Lipid 70 Mean	*p*	FDR
Bacteroidetes	*Rikenellaceae*	*Alistipes*	2.67 ± 0.12	2.03 ± 0.11	0.0003	0.0059
Firmicutes	*Oscillospiraceae*	*Oscillibacter*	1.20 ± 0.04	1.68 ± 0.10	0.002	0.018
Firmicutes	*Lachnospiraceae*	*Butyrivibrio*	0.80 ± 0.07	0.34 ± 0.04	0.002	0.017
Firmicutes	*Clostridiaceae*	unclassified	0.58 ± 0.02	0.71 ± 0.03	0.002	0.017
Firmicutes	*Ruminococcaceae*	*Anaerotruncus*	0.25 ± 0.01	0.36 ± 0.04	0.0011	0.015
Firmicutes	*Erysipelotrichaceae*	unclassified	0.14 ± 0.01	0.20 ± 0.02	0.002	0.017
Firmicutes	*Streptococcaceae*	Lactococcus	0.13 ± 0.01	0.06 ± 0.01	<0.001	0.006
Firmicutes	*Lachnospiraceae*	*Enterocloster*	0.13 ± 0.01	0.34 ± 0.04	<0.001	<0.001
Firmicutes	*Oscillospiraceae*	unclassified	0.09 ± 0.01	0.12 ± 0.01	<0.001	0.006
Proteobacteria	*Enterobacteriaceae*	*Klebsiella*	0.06 ± 0.01	0.04 ± 0.01	0.004	0.032
Firmicutes	*Lachnospiraceae*	*Schaedlerella*	0.03 ± 0.01	0.09 ± 0.01	<0.001	0.006
Firmicutes	*Streptococcaceae*	unclassified	0.02 ± 0.01	0.04 ± 0.01	0.001	0.01
Firmicutes	*Erysipelotrichaceae*	*Faecalibaculum*	0.01 ± 0.01	0.10 ± 0.02	<0.001	<0.001

**Table 3 ijms-24-10433-t003:** Functional genes annotated to the SEED database at level 4 with significant differential abundances (FDR < 0.05, |logFC| > 0.263) between rats provided with water (non-probiotic) or water supplemented with the *Lacticaseibacillus rhamnosus* strain HN001 (LactoB HN001™). A positive logFC value indicates higher abundance in the HN group, while a negative logFC value indicates higher abundance in the non-probiotic group.

Level 1	Level 2	Level 3	Level 4	logFC	FDR
Cell Envelope	Cell Envelope, Capsule and Slime layer	Capsule and Slime layer	Colanic acid synthesis	1.80	0.0001
Cell Envelope	Cell Envelope, Capsule and Slime layer	Capsule and Slime layer	Rcs two-component regulator of capsule synthesis	1.35	0.0242
Energy	Respiration	Electron-accepting reactions	Cytochrome bo ubiquinol oxidase	1.43	0.0242
Metabolism	Amino acids and Derivatives	Lysine, Threonine, Methionine, and Cysteine	S-methylmethionine	1.12	0.0393
Metabolism	Carbohydrates	Carboxylic acids	Alpha-acetolactate operon	1.01	0.0369
Metabolism	Carbohydrates	Monosaccharides	Hexose phosphate transport system	1.20	0.0485
Metabolism	Fatty acids, Lipids, and Isoprenoids	Fatty acids	Phospholipid and Fatty acid biosynthesis-related cluster	1.15	0.0319
Metabolism	Iron Acquisition and Metabolism	Siderophores	Siderophore Aerobactin	1.04	0.0167
Metabolism	Iron Acquisition and Metabolism	Siderophores	Siderophore Enterobactin	1.25	0.0303
Stress Response, Defense, Virulence	Stress Response, Defense and Virulence	Resistance to antibiotics and toxic compounds	Fusaric acid-resistance cluster	2.19	0.0014
Stress Response, Defense, Virulence	Stress Response, Defense and Virulence	Stress Response: Osmotic stress	Osmotic-stress cluster	0.98	0.0396
Stress Response, Defense, Virulence	Stress Response, Defense and Virulence	Stress Response	Rcn nickel and cobalt homeostasis system	1.39	0.0167

## Data Availability

Data may be made available upon reasonable request.

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
