# Peer review of "A Diet Enriched with Lacticaseibacillus rhamnosus HN001 and Milk Fat Globule Membrane Alters the Gut Microbiota and Decreases Amygdala GABA a Receptor Expression in Stress-Sensitive Rats"

_ijms, 2023, doi:10.3390/ijms241310433_

Round 1

Reviewer 1 Report

The authors investigated behavioural, physiological and gut microbiome responses to Lacticaseibacillus rhamnosus strain HN001 (LactoB HN001), which has been shown to reduce postpartum anxiety and depression. They examined behaviour in the open field, elevated plus maze and novel object recognition tests, in conjunction with expression of host genes in neuro-signaling pathways, and brain lipidomics. Although the literature reports a large casuistry, the casuistry with the above-mentioned strain has not been evaluated so far. If the mouse strains used do not make the typifications clear, it would be legitimate for them to be made in duplicate with a similar breed in behavioral terms. As for age, it is good to review the neuronal plasticity of young rats. The results demonstrate that a probiotic (L. rhamnosus HN001) with postpartum anxiolytic and antidepressive properties was able to increase expression of a key excitation modulatory glutamate receptor in the amygdala. These changes coincided with changes in the microbiome. In particulary the results demonstrate synergistic physiological changes exerted by the L. rhamnosus HN001 + Lipid 70 ingredient combination in reducing expression of GABAA receptor genes, suggesting neurochemical changes in key neural pathways involved in fear, anxiety and depression.

Author Response

The reviewer notes that the lack of a behavioural effect of L. rhamnosus HN001 in the Wistar Kyoto rat strain might well be due to strain dependence and suggests this be duplicated in another rat strain. We agree and are taking this into account for our next study examine the behavioural effects of foods in the Sprague Dawley rat strain under stress from social isolation. The reviewer also notes the importance of neuronal plasticity in young rats, and indeed our next study will use younger rats.

Reviewer 2 Report

1. line 22, please list the abbreviation for short chain fatty acid, the same as in line 94.

2. line 26-27, genes should be italic.

3. line 57, “[8,9] [10] [11] [12] [13];” changed as “[8-13].”, please check the full text.

4. The authors should consider that the model of this study is a 2 x 2 factorial design, thus the statistical model could be designed including the two factors (Lacticaseibacillus rhamnosus strain HN001 yes or no, and Lipid 70 yes or no) and their interaction.

5. How did the authors construct a “stress-sensitive rat” model? Please describe briefly in the Materials and Methods

1. line 22, please list the abbreviation for short chain fatty acid, the same as in line 94.

2. line 26-27, genes should be italic.

3. line 57, “[8,9] [10] [11] [12] [13];” changed as “[8-13].”, please check the full text.

4. The authors should consider that the model of this study is a 2 x 2 factorial design, thus the statistical model could be designed including the two factors (Lacticaseibacillus rhamnosus strain HN001 yes or no, and Lipid 70 yes or no) and their interaction.

5. How did the authors construct a “stress-sensitive rat” model? Please describe briefly in the Materials and Methods

Author Response

Comments and Suggestions for Authors

  1. line 22, please list the abbreviation for short chain fatty acid, the same as in line 94.

Added to correct.

  1. line 26-27, genes should be italic.

These have been edited to correct.

  1. line 57, “[8,9] [10] [11] [12] [13];” changed as “[8-13].”, please check the full text.

This has been edited to correct.

  1. The authors should consider that the model of this study is a 2 x 2 factorial design, thus the statistical model could be designed including the two factors (Lacticaseibacillus rhamnosus strain HN001 yes or no, and Lipid 70 yes or no) and their interaction.

In the paper we discuss the model with a single predictor variable, Treatment, which has four levels Control, HN, MF and MF+HN. The 2x2 factorial model proposed by the reviewer suggests converting this categorical variable into its components related to the two treatments. That is, converting this variable into two dummy variables, HN001 and Lipid70 instead, where the values of the two dummy variables are coded either zero (0) or one (1) to represent whether an animal in that group received any of the two ingredients. For example, animals in the control group would be coded as 0 and 0 for both variables, while animals in the MF + HN would be coded as 1 in both cases.

We note that when analysing the data in R, the predictor variable Treatment is automatically recoded in this way. By default, R uses reference level coding, that is, the beta coefficients would be telling us what the expected shift/change in the response when contrasted with the baseline group, which in this case would have been the control group.

In summary, we have considered the 2x2 model and get the same result with that as with the single predictor (diet) model used.

  1. How did the authors construct a “stress-sensitive rat” model? Please describe briefly in the Materials and Methods

Because the Wistar Kyoto rat strain is predisposed to stress in terms of its basal physiology (Ref 25) these animals perform poorly in behavioural tests of acute stress (ref 26 and 27) thus no further stressor as such was required for this “stress sensitive” model.

Reviewer 3 Report

In this manuscript from Dalziel et al., the authors sought to test the combinatorial effect of probiotic supplements: L. rhamnosus HN001 and Lipid 70 on anxiety and depression in the Wistar Kyoto rat strain that shows a genetic predisposition to anxiety and depression. Although both diets alone or in combination did not produce any observable behavioral effects in tests of anxiety, the combination acted synergistically and produced gene expression changes in the GABA signaling pathway in the amygdala. The authors have aptly discussed the contradictory results and limitations of the study.

Strengths:

11. Use of a well-established animal model of anxiety.

E2. Expression analysis using NanoString technology.

33. Mass spectrometry for lipidomics and fatty acid analysis.

44. Use of appropriate statistical methods.

No major weakness was noted.

Some minor comments:

11)      A, B, and C panels in Figure 1 can be consolidated into one figure.

22)      Abbreviations are usually defined at first use. OFT, EPM, NOR in line 106 are defined much later in the methods section.

33)      Table 3 can be moved to the supplementary materials.

44)      What is Sparse Partial Least Squares Canonical analysis?

Author Response

Some minor comments:

11)      A, B, and C panels in Figure 1 can be consolidated into one figure.

Thank you for this suggestion. These plots represent either different treatment groups or brain regions so we are reluctant to group them together. We have reworded the legend for clarity.

22)      Abbreviations are usually defined at first use. OFT, EPM, NOR in line 106 are defined much later in the methods section.

Underlined test has been added:

“No behavioural differences were detected due to treatment in any of the assessment measures across: the open field test (OFT), indicative of anxiety and exploration; the elevated plus maze (EPM) test, indicative of anxiety; and the novel object recognition (NOR) test for recognition memory.”

33)      Table 3 can be moved to the supplementary materials.

This has been moved to “Supplementary Table 3”.

44)      What is Sparse Partial Least Squares Canonical analysis?

Partial least squares (PLS) regression is a multivariate method used to model the relationship between two matrices, X and Y. The fitted model is found by finding the multidimensional direction in the X space that explains the maximum multidimensional variance direction in the Y space. The sparse version of PLS involves including penalties on the weighting of the original variables that are used to construct the new latent variables. The sparse version of this method was chosen because it helps with the interpretability of the regression model when applying the PLS on data of high dimensionality.

Round 2

Reviewer 1 Report

I confirm the previous review. therefore the work is suitable for immediate publication.

Reviewer 2 Report

This manuscript can be accepted.